# Age-dependent appearance of SARS-CoV-2 entry sites in mouse chemosensory systems reflects COVID-19 anosmia-ageusia symptoms

Julien Brechbühl [1], Ana Catarina Lopes[1], Dean Wood[1], Sofiane Bouteiller[1], Aurélie de Vallière[1], Chantal Verdumo[1] & Marie-Christine Broillet [1]✉

COVID-19 pandemic has given rise to a collective scientific effort to study its viral causing agent SARS-CoV-2. Research is focusing in particular on its infection mechanisms and on the associated-disease symptoms. Interestingly, this environmental pathogen directly affects the human chemosensory systems leading to anosmia and ageusia. Evidence for the presence of the cellular entry sites of the virus, the ACE2/TMPRSS2 proteins, has been reported in non-chemosensory cells in the rodent's nose and mouth, missing a direct correlation between the symptoms reported in patients and the observed direct viral infection in human sensory cells. Here, mapping the gene and protein expression of ACE2/TMPRSS2 in the mouse olfactory and gustatory cells, we precisely identify the virus target cells to be of basal and sensory origin and reveal the age-dependent appearance of viral entry-sites. Our results propose an alternative interpretation of the human viral-induced sensory symptoms and give investigative perspectives on animal models.

[1] Department of Biomedical Sciences, Faculty of Biology and Medicine, University of Lausanne, Lausanne, Switzerland. ✉email: mbroille@unil.ch

The Corona Virus Disease 2019 (COVID-19) has federated worldwide scientific efforts for understanding the viral epidemiological mechanisms of the coronavirus 2 (SARS-CoV-2) that causes this severe acute respiratory syndrome. In humans, the viral syndrome is characterized by an increased mortality rate in aged and/or comorbidity patients associated with the upper respiratory infection symptoms, such as severe respiratory distress[1–3]. In addition to its major impact, COVID-19 is associated by its direct alteration of human olfaction and gustation, in absence of substantial nasal inflammation or coryzal signs, resulting to anosmia and ageusia in up to 77% of the patients[4–7]. While these sensory symptoms are well established and intensely affect everyday behaviors[8,9], the precise related mechanisms remain elusive[10].

The target cells of the virus share a molecular signature: the concomitant cellular expression of the angiotensin-converting enzyme 2 (ACE2) and of its facilitating transmembrane serine protease 2 (TMPRSS2), which plays a crucial role for the interaction of viral spike proteins with the host cell[11–13]. Paradoxically, these entry sites seem to be lacking in sensory cells[14–18], while a direct SARS-CoV-2 contamination has been observed both in humans and rodents[19,20], requesting further investigations to explain the sensory-associated symptoms[21–24]. Therefore, the characterization of the animal model is necessary prior to its use to understand the causality underling the viral-induced sensory symptoms.

The use of mice is indeed limited for epidemiological studies due to their absence of hands, which, with aerosols, are the foremost passages of interindividual viral transmission[25], as well as their published lack of SARS-CoV-2 ACE2-spike protein affinity[26,27]. Nevertheless, the ease of production of genetically modified mice and their scientific availability, as well as their well-studied and specialized chemosensory systems[28–30], make them a valuable ally for the development of potential prophylactic and protective treatments related to these sensory symptoms.

Thus, we aimed here at characterizing the potential viral entry sites across mouse sensory systems. We found SARS-CoV-2 entry cells to be of different origins depending on the sensory systems. In summary, the virus could target cells involved in tissue regulation such as the supporting cells of the olfactory receptor neurons and the regenerative basal cells but also, specifically, the chemosensory cells of both gustatory and olfactory systems. We finally revealed that the emergence of viral entry sites in sensory and basal cells only occurs with age, which could explain both, the observed COVID-19 long-lasting effects and the age-dependent sensory-symptomatology in human.

## Results

**SARS-CoV-2 entry genes are differentially expressed in the mouse sensory systems.** Focusing on the mouse, where chemosensing takes place in different sensory systems (Fig. 1a), we took advantage of our previous studies[31–35] to first assess their overall expression of the major SARS-CoV-2 entry sites. Interestingly, we found a differential expression of the *Ace2* and *Tmprss2* transcripts (Fig. 1b, c and Supplementary Fig. 1). *Ace2* is strongly expressed in a specific area of the main olfactory epithelium (MOE), the dorsal part (MOE$_D$) which is directly exposed to the environment and specialized in sensing volatile chemical cues[14,15]. Moreover, we observed a previously unreported expression of *Ace2*, in the most rostral sensory subsystem, the Grueneberg ganglion (GG), mostly implicated in volatile danger cues detection[34–36] as well as in the different taste papillae, the fungiform (Fu), the foliate (Fo) and the circumvallate (CV) taste papillae, all involved in water-soluble tastant perception[29]. Interestingly, this specific pattern of *Ace2* expression seems to be correlated with the mode of viral dissemination (volatile suspensions of viral droplets[25]), as only a limited expression of *Ace2* is found in the vomeronasal organ (VNO) and in the septal organ of Masera (SO), considered to be implicated in pheromonal and retronasal communications via indirect and only limited access to the environment[30]. Concerning the associated facilitating cleavage protease, we found that the *Tmprss2* transcript was expressed in all olfactory subsystems and to a limited extent at the taste Fu, Fo and CV level (Fig. 1b, c and Supplementary Fig. 1). Thus, this apparent disparity of expression requires further investigations to identify the precise cells expressing these entry sites in the different mouse subsystems.

**SARS-CoV-2 entry cells in the MOE$_D$ are of non-neuronal and multipotent types.** Taking advantage of a genetically modified mouse model in which, the olfactory marker protein[37] (OMP) drives the expression of the green fluorescent protein (GFP) in all mature olfactory neurons[38,39], we first examined the MOE$_D$ sensory epithelium (Fig. 2) which is continuously exposed to inhaled air (Fig. 2a). Remarkably, we found by immunohistochemical stainings that the ACE2 protein was not only expressed at the apical surface and in the Bowman's glands[14–16,21] but also in the basal layer (Fig. 2b). Moreover, we found that, in addition to being irregularly localized in different regions of the neuroepithelium[15] (Supplementary Fig. 2a), TMPRSS2 was mostly co-expressed with ACE2 in these OMP-negative basal cells (Fig. 2c, d). Focusing on the identification of these ACE2-expressing cells (Fig. 3), we first confirmed that the cytokeratin 18 (CK18)-positive sustentacular cells[16,21], operating as supporting cells for the olfactory neurons, were indeed harboring, at their luminal surface, the observed ACE2 protein (Fig. 3a). Moreover, we found ACE2 apical expression to be localized below the olfactory sensory cilia expressing the cyclic nucleotide-gated channel alpha 2 subunits (CNGA2; Supplementary Fig. 2b), supporting its previously reported sustentacular microvilli affiliation[14,15]. We next established the multipotency characteristics of the ACE2-expressing basal cells, as they also expressed two proteins specific to their identity of stem cells, the perinuclear cytokeratin 5 (CK5; Fig. 3b) and, predominantly, the nuclear transcription factor sex determining region Y-box 2 (SOX2; Fig. 3c), markers of horizontal basal cells (HBCs) or of the pear-shaped globose basal cells (GBCs) respectively. These two populations of basal cells are both involved in the regeneration of the neuroepithelium by acting as short- and long-lasting reservoir cells[14,40,41]. Taken together, we confirmed our initial RT-PCR and qRT-PCR results (Fig. 1b, c) by precisely profiling the ACE2 and TMPRSS2 expression in the MOE$_D$ neuroepithelium. We found that the cell candidates for viral entry are ACE2-expressing cells of non-neuronal and multipotent origin.

**SARS-CoV-2 entry cells in taste papillae are mature sensory cells.** Wondering about the apparent limited level of *Tmprss2* transcript in the different taste papillae (Fig. 1b, c), we next decided to exploit our histological methodology (Figs. 2, 3) to precisely localize the potential SARS-CoV-2 entry sites in sensory cells of the Fu, Fo and CV taste buds (Fig. 4 and Supplementary Figs. 3 and 4). For that, we first used G$_\alpha$ gustucin (GUST) as a marker of sensory cells in the tongue[32], to identify taste buds in close contact with the oral cavity (Fig. 4a, Supplementary Figs. 3a and 4a). Surprisingly, using a double immunostaining approach, we distinctly observed ACE2 expression, not only in apical keratinocytes[17,42], but also in sensory cells (Fig. 4b, Supplementary Figs. 3a and 4a). Moreover, ACE2 is predominantly expressed in the microvilli of sensory cells, gathered in the so-called taste pore, which are directly exposed to substances entering the oral cavity[29], such as viral droplets (Fig. 4b,

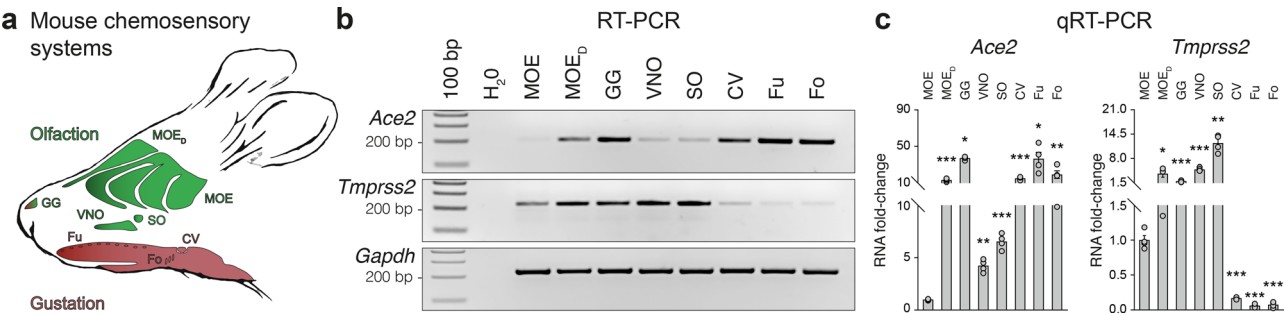

**Fig. 1 Differential expression of the SARS-CoV-2 entry site transcripts in the mouse chemosensory systems. a** Schematic representation of a mouse head with its chemosensory systems. Olfactory neurons are distributed in different olfactory subsystems: the main olfactory epithelium (MOE) and its dorsal region (MOE$_D$), the vomeronasal organ (VNO) and the septal organ of Masera (SO). Chemosensory cells for gustation are found in different tongue regions, grouped in taste buds which are presented by taste papilla such as the fungiform (Fu), the foliate (Fo) and the circumvallate papillae (CV). The Grueneberg ganglion (GG), the most rostral olfactory subsystems, displays both olfactory and gustatory properties. Gene expression profiles of the SARS-CoV-2 entry sites: *Ace2* and *Tmprss2* in the different mouse chemosensory systems performed by RT-PCR (**b**) and quantified by qRT-PCR (**c**). *Gapdh* is used as a reporter gene and $H_2O$ as a negative control of transcript expression. Samples for gene expression profiles are obtained from 5 to 10 heterozygous OMP-GFP mice of 4–11 months old. Data are expressed as an RNA fold-change relatively to the MOE and represented as mean ± SEM with aligned dot plots for $n = 4$ individual sample values (**c**). For comparisons between conditions, two-tailed Student's *t*-tests or Mann–Whitney tests are used, $*p < 0.05$, $**p < 0.01$, $***p < 0.001$. Ladder of 100 base pairs (bp, (**b**)).

**Fig. 2 Expression profiles of the ACE2 and TMPRSS2 proteins in the MOE$_D$.** Immunohistochemical investigations on the MOE$_D$ for SARS-CoV-2 entry sites (ACE2, in pink; TMPRSS2, in red). Here, the olfactory marker protein (OMP, in green) allows the precise localization of the mature olfactory neurons. **a** Large view of OMP+ neurons of the MOE$_D$. Chemosensory cells are in direct contact with the nasal cavity. **b** Enlarged view of ((**a**), dashed rectangle), showing expression of ACE2 in the apical surface, Bowman's gland and basal layer. A zoom in view of the apical surface is shown. **c** Co-expression profiles of TMPRSS2 and ACE2 in OMP- basal cells (white arrowhead, highlighted with zoom in view). **d** Control experiment (Ctrl neg) illustrating the absence of fluorescent background expression. Nuclei are counterstained with Dapi (DAPI, in blue). Representative protein expression profiles obtained from heterozygous OMP-GFP mice of 20 (**a**, **b**) and 6 months old (**c**, **d**). Scale bars are 50 µm (**a**), 15 µm (**b**), and 10 µm (**c**, **d**).

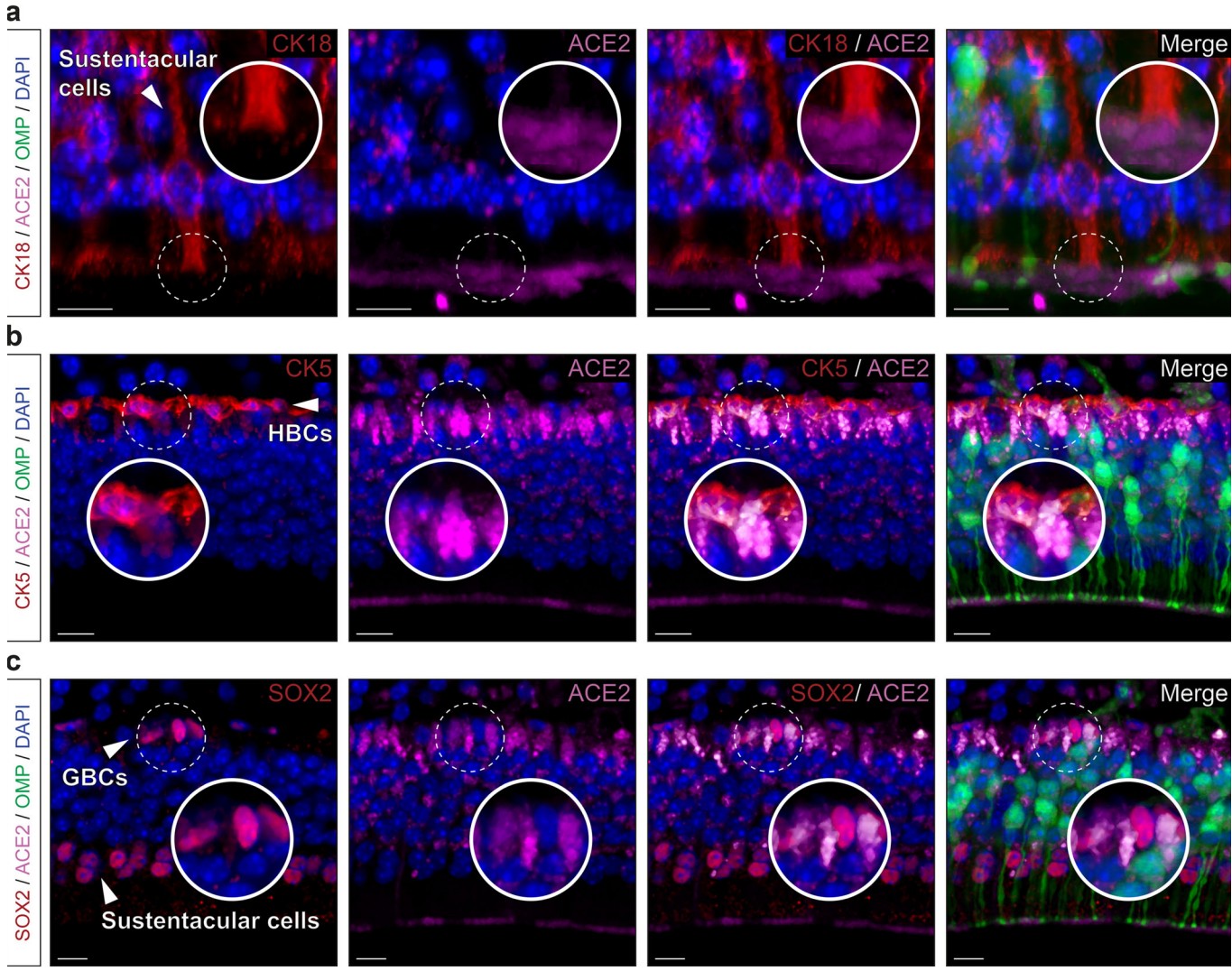

**Fig. 3 Non-neuronal and immature olfactory cell localization of ACE2 in MOE$_D$.** Cellular profile of ACE2 expression in the MOE$_D$ by immunohistochemistry. **a** Co-expression of ACE2 (in pink) with apical CK18+ sustentacular cells (CK18, in red). **b**, **c** Basal co-expression of ACE2 with CK5 + HBCs (CK5, in red; (**b**)) and SOX2 + GBCs (SOX2, in red; (**c**)). Nuclei of sustentacular cells (**c**) also expressed SOX2. Colocalization between red and pink signals is highlighted in light gray (**a–c**). Nuclei are counterstained with Dapi (DAPI, in blue). Horizontal basal cells (HBCs, (**b**)), globose basal cells (GBCs, (**c**)). Representative protein expression profile obtained from heterozygous OMP-GFP mice of 9 (**a**, **b**) and 7 months old (**c**). Scale bars are 5 µm (**a**, **b**) and 10 µm (**c**).

Supplementary Figs. 3a and 4a). Furthermore, we observed a restricted localization of TMPRSS2 in the taste pore region, strikingly co-expressed with ACE2 (Fig. 4c, d, Supplementary Figs. 3b and 4b) which might also explain the observed low-level signal for the *Tmprss2* transcript (Fig. 1b, c). We next confirmed the mature status of these ACE2-expressing sensory cells as they preferentially expressed the CK18 marker[43] (Fig. 5a). Moreover, and contrary to the MOE$_D$, we found only sporadic ACE2 expression in basal CK5-positive cells (Fig. 5b) and SOX2-positive cells (Fig. 5c, d). In summary, we identified here the co-expression of the main SARS-CoV-2 entry sites in mature taste sensory cells.

**SARS-CoV-2 entry cells in the GG are sensory neurons.** In the mouse, there is an ancestral sensory subsystem, the GG (Fig. 6), which has the particularity of displaying both olfactory and gustatory traits[31,32,44]. Morphologically, GG neurons are covered with a keratinized epithelium permeable to volatile water-soluble molecules coming from the nasal cavity[35] (Fig. 6a). Molecularly, it expresses both main sensory markers, OMP and GUST[32] and uses common

sensory signals such as olfactory and taste receptors[32] to respond to both danger-associated odorants and tastants[32,35,36]. Performing histological approaches, we first noticed that the apical keratinocyte cell layer was, as in the different taste papillae, positive for ACE2 expression (Fig. 6b). But more surprising, we next found ACE2 expression in the GG cells (Fig. 6b). Interestingly and contrary to the MOE$_D$, this expression was distinctly restricted to the sensory GG neurons and not found in the supporting S100 calcium-binding protein β (S100B)-expressing glial cells[35] (Supplementary Fig. 5a). Remarkably, we furthermore localized TMPRSS2 both in apical keratinocytes and in GG neurons indicating that the two main SARS-CoV-2 entry sites are indeed found in the GG sensory subsystem (Fig. 6c, d). Looking for a precise characterization of ACE2-expressing cells in the GG, we next discovered that OMP-positive GG neurons were also CK18-positive (Fig. 7a) giving the GG a double cellular affiliation (neuronal and fibroblastic) that also highlights the discovery of CK18 as a reliable marker of ACE2-expressing cells, the putative target cells of viral infection[45]. We next observed that in the apical keratinocyte layer of the GG, ACE2-expressing cells are mostly mature as they rarely co-expressed CK5

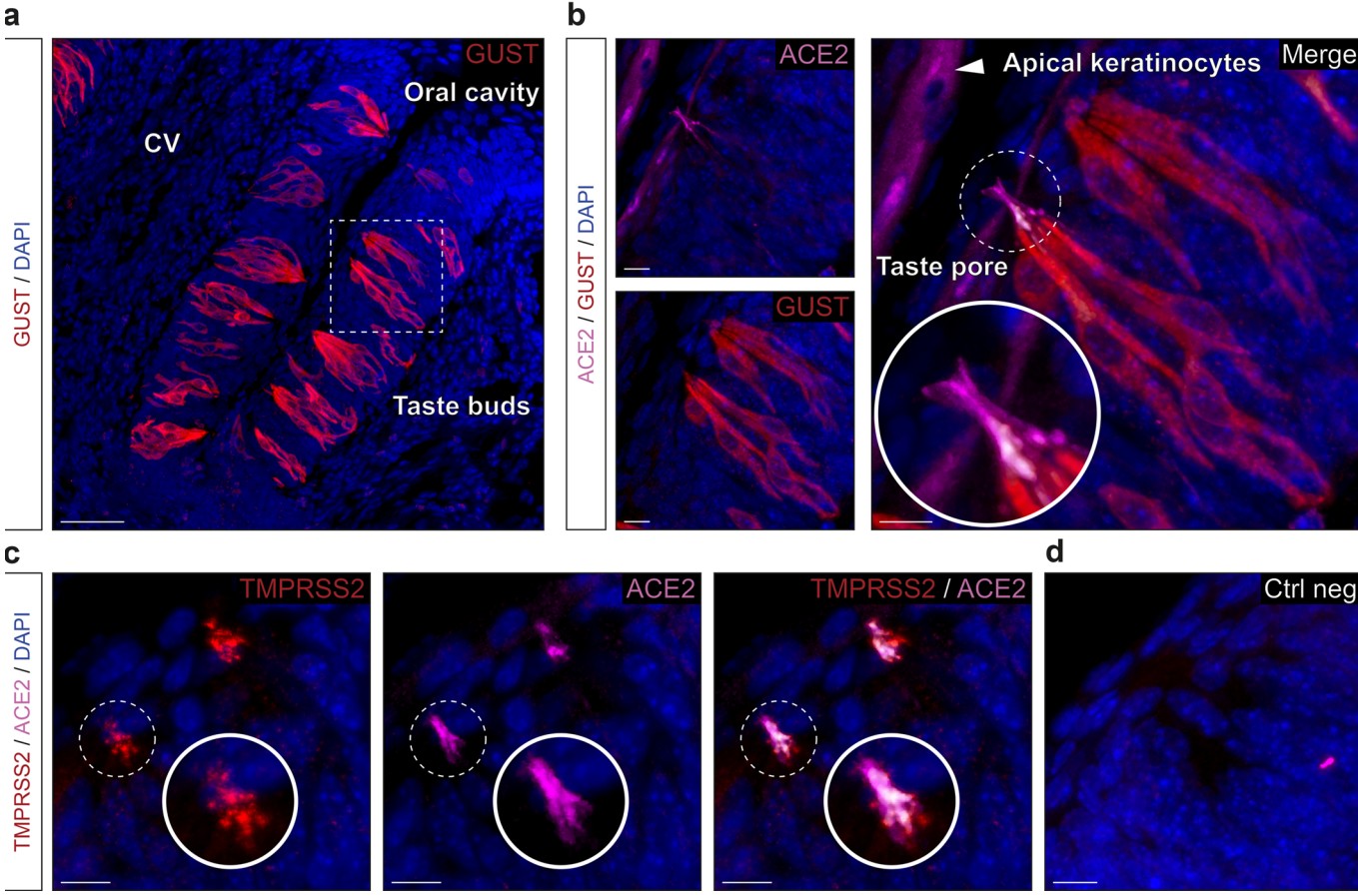

**Fig. 4 Expression profile of the ACE2 and TMPRSS2 proteins in the taste circumvallate papilla.** Immunohistochemical investigations on the taste buds for SARS-CoV-2 entry sites in circumvallate papillae. **a, b** Here, the gustducin marker protein (GUST; in red) allows the localization of the gustatory sensory cells. **a** Large view of GUST+ sensory cells of the circumvallate papilla (CV). **b** Enlarged view of ((**a**), dashed rectangle), showing expression of ACE2 (in pink) in the apical keratinocytes and in the microvilli of the chemosensory cells forming the taste pores, which are in direct contact with the oral cavity. A zoom in view of the microvilli of a taste pore is shown. **c** Co-expression profile of TMPRSS2 (in red) and ACE2 in the taste pores (highlighted with a zoom in view). **d** Control experiment (Ctrl neg) illustrating the absence of fluorescent background expression. Colocalization between red and pink signals is highlighted in light gray (**a–d**). Nuclei are counterstained with Dapi (DAPI, in blue). Representative protein expression profiles were obtained from heterozygous OMP-GFP mice of 6 (**a**, **b**), 5 (**c**), and 7 months old (**d**). Scale bars are 50 µm (**a**), 10 µm (**b–d**).

and SOX2 (Supplementary Fig. 5b, c), a characteristic shared with taste tissue (Fig. 5). Interestingly, and as previously shown, no regeneration happened throughout the lifespan in the GG subsystem[31,46] as basal cells are absent which was further confirmed here by a lack of SOX2 staining (Supplementary Fig. 5c). However, we observed in GG neurons, a punctiform-like staining for ACE2 and CK5 (Supplementary Fig. 5b) indicating a co-expression in a precise cytoskeleton structure, as for the taste buds (Fig. 4b, Supplementary Figs. 3a and 4a). We found that these cytoskeleton regions were associated to GG primary cilia (Fig. 7b, c), an organelle implicated in chemosensing[33,36]. Thanks to the particulate guanylyl cyclase G (pGCG; Fig. 7b), a marker of the axonemes of the GG-cilia[33], we observed that ACE2 was located in the so-called basal body structures (Fig. 7c), where it was also co-expressed with the gamma-tubulin marker[35] (γ-TUB; Fig. 7c). In summary, we found that the GG sensory subsystem possesses both the viral target cells and the specific protein profiles of expression displayed by both the olfactory and taste systems. Moreover, we bring here evidences of the presence of viral entry sites in a particular population of chemosensory cells, the olfactory GG neurons.

**SARS-CoV-2 entry sites expression in mouse chemosensory systems is age-dependent.** During the course of our experiments, we noticed interindividual variations of ACE2 expression not

only in its intensity but also in its general arrangement (*e.g.*, apical versus basal ACE2 expression in the MOE_D; (Fig. 2b, c). Moreover, this observation was not only done for the MOE_D (Figs. 2, 3), but also for the taste buds of the CV papillae (Figs. 4, 5) as well as for the GG (Figs. 6, 7). We considered a potential age-dependent correlation[47–50] and thus we next challenged the expression of *Ace2* and *Tmprss2* transcripts through the different chemosensory subsystems using mice of chosen ages (Fig. 8 and Supplementary Fig. 6). Using both RT-PCR and qRT-PCR experiments, we confirmed our assumption at the RNA level as we observed and quantified a significant increase of both *Ace2* and *Tmprss2* transcripts with age in the MOE_D (Fig. 8a, d and Supplementary Fig. 6a, d), in the CV (Fig. 8b, e and Supplementary Fig. 6b, e) and in the GG (Fig. 8c, f and Supplementary Fig. 6c, f). As this age-dependence was particularly striking for *Ace2*, we next focused on its expression at the protein level (Fig. 9 and Supplementary Fig. 7). After a first validation of this age-dependent trend in a series of western-blot quantifications (Supplementary Fig. 7a–c), we subsequently undertook a precise immunohistochemical localizations of ACE2 expression in the different chemosensory subsystems (Fig. 9). Accordingly, we observed an intense variation of its expression in the MOE_D (Fig. 9a). Indeed, in young mice, this expression was restricted to the apical region, and then gradually intensifies with age (Fig. 9a).

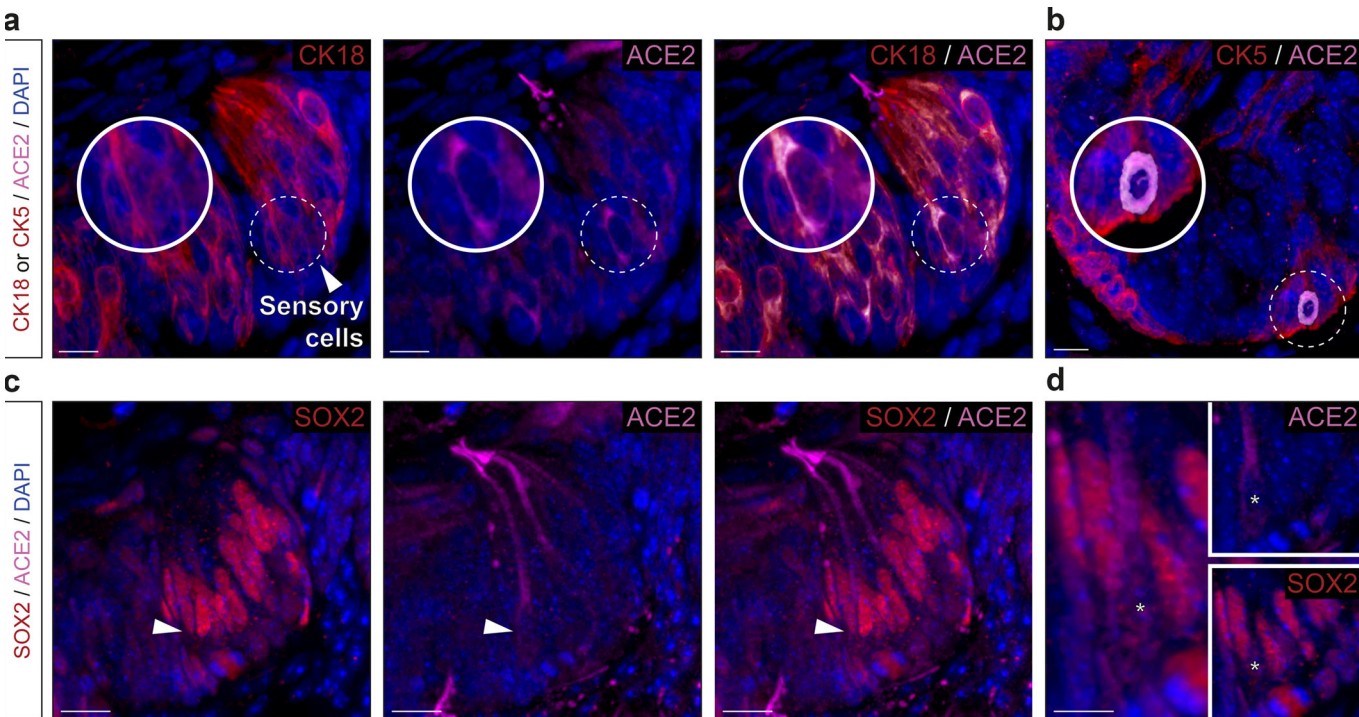

**Fig. 5 Precise protein expression profile of ACE2 in CV taste buds reveals its presence in mature sensory cells.** Cellular profile of ACE2 expression in the CV taste buds by immunohistochemistry. **a** Co-expression of ACE2 (in pink) with CK18+ sensory cells (CK18, in red). Enlarged view (dashed white circle) shows episodic expression of ACE2 in sensory cell soma. **b** Limited basal co-expression of ACE2 in immature CK5+ cells (CK5, in red). **c**, **d** Somatic expression of ACE2 is associated with nuclear expression of SOX2+ (SOX2, in red). **d** Enlarged view of a nuclear expression of SOX2 (indicated by white asterisks) in a ACE2+ sensory cell (white arrowhead in (**c**)). Colocalization between red and pink signals is highlighted in light gray (**a–d**). Nuclei are counterstained with Dapi (DAPI, in blue). Representative protein expression profile obtained from heterozygous OMP-GFP mice of 11 (**a**, **b**) and 13 months old (**c**, **d**). Scale bars are 10 μm (**a–c**) and 5 μm (**d**).

At the basal level, it appears to be absent first and only shows up in adult mice (Fig. 9a). Moreover, these variations were directly associated with an increase in ACE2 expression in existing cells. The sustentacular and basal cells are present at all ages of interest (Fig. 9b). In the sensory cells of the CV, ACE2 appears to be absent in young mice and only emerges with age (Fig. 9c). Interestingly, ACE2 expression initiates in the microvilli, at around 5 months of age, and then gradually extents throughout the cell body with aging (Fig. 9c). Concerning the GG, we first noticed that its apical keratinocyte cell layer constitutively expressed ACE2, while its expression was distinctly age-dependent in the sensory neurons (Fig. 9d). Indeed, and as for the sensory cells of the CV (Fig. 9c), a subcellular expression was first observed which spread into the soma with time (Fig. 9d). Overall, we demonstrated here a striking age-dependent increase of SARS-CoV-2 entry sites expression, in particular for ACE2 across mouse chemosensory systems.

## Discussion
Our ability to perceive and interact with our environment is directly linked to our senses. From an evolutionary point of view, our sensory performance is often associated with the way we communicate with our surroundings. In humans, although olfaction and taste have lost importance in comparison with mice, they remain essential not only to our well-being but also to protect us from dangers such as intoxication[10]. Their disorders are correlated with our general and mental health and can be early indicators of central nervous system impairments[51,52]. Our sensory systems have developed a variety of protective mechanisms to guard us against environmental attacks such as the presence of protective keratinous epithelia, the combined action of

mucus and cilia of sustentacular cells, ensuring the drainage of environmental microorganisms via mucociliary fluid clearance[53] or even the presence of regenerative basal cells[40,41]. SARS-CoV-2 seems to have found a mechanism to thwart our sensory defenses. Indeed, we have shown here that, in mice, these very protective cells express viral entry sites. Although the direct action by the virus on these cells remains to be demonstrated, the underlying inflammatory[22,24,54] or cytopathic destruction mechanisms[23] would strongly impact the senses of smell and taste. It therefore seems obvious that their targeted impairment can directly contribute to long-term anosmia and ageusia[7,55]. Moreover, we have demonstrated here that the sensory cells themselves also express the SARS-CoV-2 entry sites which could contribute to both the sensory symptoms observed on the short term[14,15,19,21] and the ability of coronaviruses to directly infect human sensory cells[20].

From a clinical point of view, anosmia and ageusia have a low prevalence in infected children and increase with the age of the COVID-19 patient[50,56], which seems to be consistent with the protein expression of ACE2 that we have observed in the mice olfactory system. Indeed, after we first confirmed the age-dependent trend made of ACE2 expression in sustentacular cells[47], we focused on the dorsal part of the MOE and found that the basal cells of the neuronal epithelium also expressed ACE2 as a function of age. Interestingly, TMPRSS2 is also expressed in these SARS-CoV-2 target cells. However, further studies, using viral inoculation for example, are still necessary to link this specific protein expression with viral sensitivity. Moreover, we also observed this age-effect in the taste system. As an absence of expression of the viral entry sites is found in the sensory cells of young mice, but an increased expression of ACE2 and TMPRSS2 appears in sensory cells when the mouse advances in age. Thus,

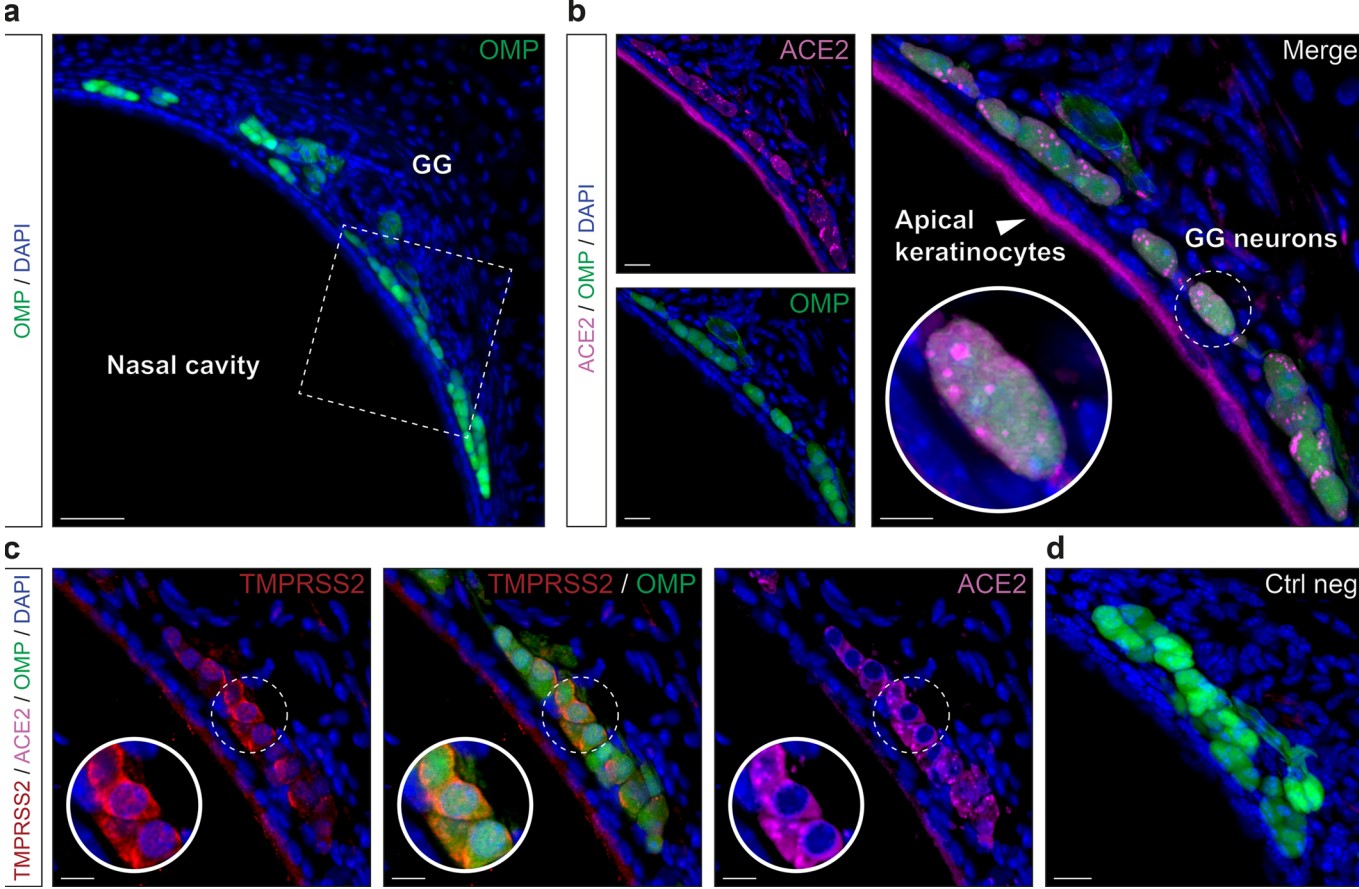

**Fig. 6 Expression profile of ACE2 and TMPRSS2 proteins in the GG.** Immunohistochemical investigations on the GG for SARS-CoV-2 entry sites (ACE2, in pink; TMPRSS2, in red). Here, the olfactory marker protein (OMP, in green) allows the precise localization of the mature olfactory neurons. **a** General view of OMP+ neurons of the GG. Chemosensory cells are separated from the nasal cavity by a keratinocyte layer. **b** Enlarged view of ((**a**), white dashed rectangle), showing expression of ACE2 in the apical keratinocytes and in GG neurons. A zoom in view of GG neurons is shown (white dashed circle). **c** Co-expression profile of TMPRSS2 and ACE2 in apical keratinocytes and OMP+ cells (zoom in view, white dashed circle). **d** Control experiment (Ctrl neg) illustrating the absence of fluorescent background expression. Nuclei are counterstained with Dapi (DAPI, in blue). Representative protein expression profile obtained from heterozygous OMP-GFP mice of 9 (**a**, **b**, and **d**) and 11 months old (**c**). Scale bars are 50 μm (**a**), 15 μm (**b**–**d**).

SARS-CoV-2 could therefore also directly target these sensory cells in aged mice expressing, for example, the human form of ACE2 (hACE2) under the mouse *Ace2* promotor[45,57] and, possibly, in aged humans[42], altering them and consequently disabling the taste sensory ability leading to ageusia symptoms. Thus, the use of animal models such as mice sensitized to the virus[45,57–60] or Syrian hamsters that are naturally compatible with SARS-CoV-2 infections[19,21,61], appears to be a promising strategy but the age of the animals has to be carefully considered and older animals should preferentially be used (from 9 months of age for mice). To reinforce the importance of this notion, previous elegant single cell transcriptomic analysis indeed reported only limited expression of *Ace2* in taste cells of young and embryonic mice[17]. We reported here a similar observation both at the RNA and protein levels in young mice and we completed this description by looking at ACE2 expression in aged mice. Thus, we found that the age of the animal directly impacts the interpretations of the results obtained in SARS-CoV-2 studies[17,42,62]. Interestingly, based on our results, additional studies performed on the lower respiratory airways could show whether a similar increased ACE2 expression, as a function of age occurs not only in respiratory ciliated cells[49,63] but also in basal cells which could therefore contribute, in association with other factors such as interferon-stimulation[64,65], to the differences in pulmonary infectiousness observed between young and older people[66].

Mice possess multiple sensory systems, separated into specialized subsystems[30], that we have now precisely characterized for their expression of the viral entry proteins. Interestingly, our study identified the GG to express the SARS-CoV-2 entry sites on its sensory neurons. We also found that GG sensory neurons expressed CK18, confirming that the cellular expression of this protein is an excellent marker for the localization of putative viral entry cells[45]. Moreover, this observation is remarkable and gives to the GG a double cellular affiliation, namely neuronal and fibroblastic by the expression of both OMP and CK18. In this period of race for collective immunity[9], we desperately need animal models to study this disease. Our results not only confirm the unique primary origin of the GG[44] but, also make it useful as a model of sensory systems to study, in ex-vivo and in-vivo preparations[35] from hACE2 mouse or Syrian hamster[33], the mechanism of viral entry into sensory cells as well as the testing of potential protective treatments against viral infection, such as intranasal drugs and/or COVID-19 vaccine candidate delivery[13,67,68]. Reinforcing this idea, it should be noted that another SARS-CoV-2 portal of entry, neuropilin-1[69,70] is also expressed in the GG[71], thus increasing its infectious susceptibility.

In summary, we have precisely characterized the SARS-CoV-2 entry cells in the multiple sensory systems of the mouse. These cells are of different origins. They are either involved in tissue

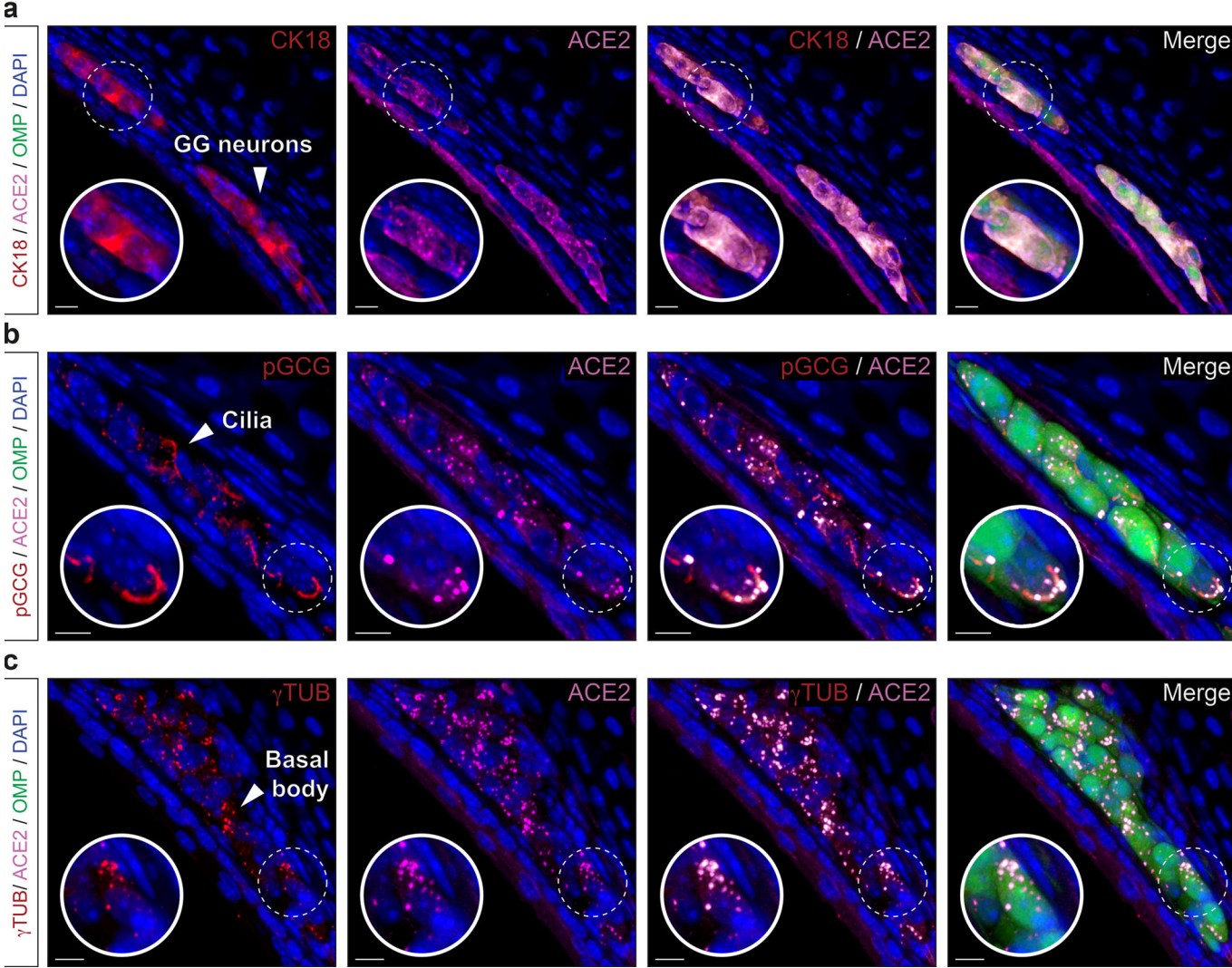

**Fig. 7 Precise protein expression profile of ACE2 in the GG reveals its neuronal type.** OMP+ cellular profile of ACE2 expression in the GG by immunohistochemistry. **a** Co-expression of ACE2 (in pink) with CK18 + GG neurons (CK18, in red). **b** ACE2 is expressed together with pGCG+ GG primary cilia (pGCG, in red). **c** Co-expression of ACE2 with γTUB+ punctiform staining in GG neurons (γTUB, in red) indicating a localization in the basal body of GG primary cilia. Colocalization between red and pink signals is highlighted in light gray (**a–c**). Nuclei are counterstained with Dapi (DAPI, in blue). Representative protein expression profile obtained from heterozygous OMP-GFP mice of 11 (**a**) and 5 months old (**b, c**). Scale bars are 10 µm (**a–c**).

function and regeneration but also, specifically, in chemosensing. We have revealed that the expression of viral entry sites in sensory cells increases with age, in this experimental model, which might be an important factor for viral infectivity studies. Thus, we have suggested a direct correlation between human sensory-symptomatology and mice SARS-CoV-2-expressing entry cells providing a putative explanation for the observed anosmia and ageusia in COVID-19 patients.

## Methods

**Animals.** C57BL/6 mice (*Mus musculus*; Janvier Labs) and heterozygous OMP-GFP mice[39], of both sexes, were used at the indicated ages. In this gene-targeted mouse strain, mature olfactory sensory neurons[37] expressed the histological reporter GFP under the control of the OMP promoter[38,39]. Mice were grouped-housed between 21 and 23 °C under a 12 h light/dark cycle with ad libitum access to food and water. Mice were euthanized by $CO_2$ inhalation and the experimental procedures were in accordance with Swiss legislation and approved by the EXPANIM committee of the Lemanique Animal Facility Network and the veterinary authority of Canton de Vaud (SCAV).

**Chemosensory epithelia isolation.** Prior to tissue isolation processes, phosphate-buffered saline (PBS; 138 mM NaCl, 2.7 mM KCl, 1.76 mM $KH_2PO_4$, and 10 mM $Na_2HPO_4$, pH 7.4), dissection tools and equipment were sterilized and RNase

removing agent (RNaseZAP™; Sigma) treatments were applied. The chemosensory epithelia were localized under the fluorescent stereomicroscope (M165 FC; Leica) thanks to the GFP expression for the olfactory subsystems[32,33]. The dorsal GFP-positive prolongation of the main olfactory epithelium ($MOE_D$), corresponding to the ethmoid turbinate 1E adherent to the nasal bone, was separated from the rest of the MOE for the analysis. The other olfactory subsystems, the VNO, the SO and the GG were delicately separated from the neighboring GFP negative tissues. The Fu, the Fo, and the CV papillae were extracted from the surface of the tongue.

**Gene expression profile by RT-PCR and qRT-PCR.** RNA purification was completed according to the instructions of the manufactured kit (RNeasy® Plus Mini kit; Qiagen). Briefly, the chemosensory epithelia obtained from 5 to 10 mice (with a comparable sex ratio) were isolated and pooled in buffer RLT Plus supplemented with b-mercaptoethanol. Homogenization was performed through high-speed tissue disruption (TissueLyser II; Qiagen) and genomic DNA was removed from lysate by using the gDNA Eliminator spin column. Centrifugation and loaded processes were performed onto RNeasy spin columns to finally elute the total RNA in 30 µl of RNase-free water. Reverse transcription (RT) was then initiated with the cDNA synthesis kit (PrimeScript™ 1st strand cDNA Synthesis Kit; Takara) using 140 ng of RNA and the random hexamers option to obtain a final volume of 20 µl of cDNA.

*RT-PCR investigations.* For reverse transcription polymerase chain reaction (RT-PCR), PCR experiments were subsequently performed using 3 µl of cDNA and 800 nM of

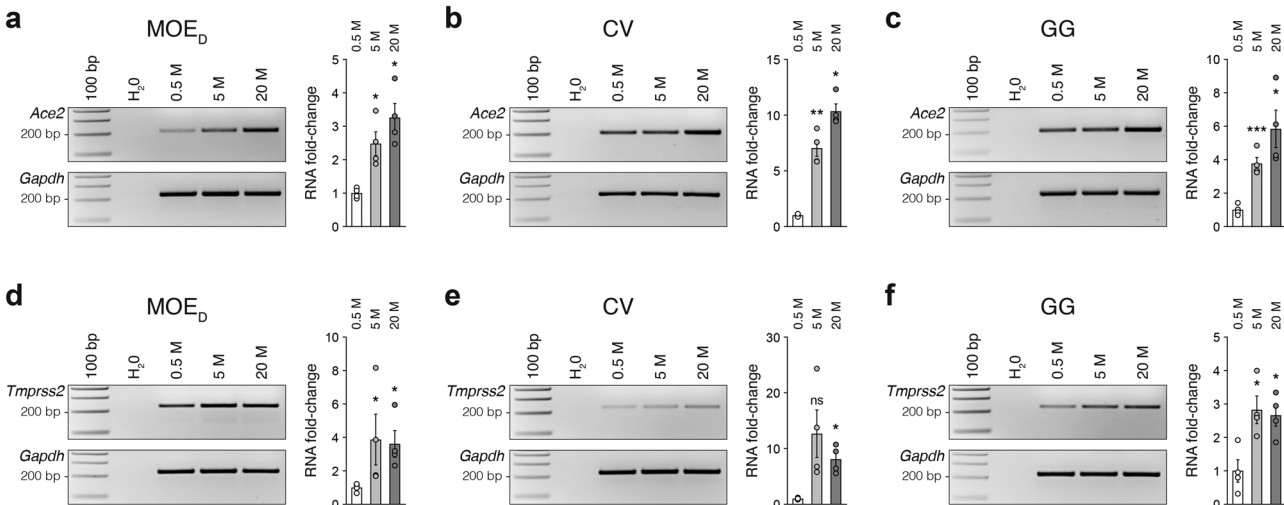

**Fig. 8 Age-dependent expression of SARS-CoV-2 entry site transcripts in the MOE$_D$, CV, and GG.** Assessment of the expression of *Ace2* and *Tmprss2* transcripts in chemosensory systems by RT-PCR (left part of panels) and qRT-PCR (right part of panels) at the indicated age in months (M). An age-dependent increase in *Ace2* (**a–c**) and in *Tmprss2* (**d–f**) is observed in the MOE$_D$ (**a**, **d**), CV (**b**, **e**), and GG (**c**, **f**). *Gapdh* is used as a reporter gene and H$_2$O as a negative control of transcript expression. Samples for gene expression profiles are obtained from 5 to 6 heterozygous OMP-GFP mice of indicated age. Data are expressed as an RNA-fold change relatively to the 0.5 M and represented as mean ± SEM with aligned dot plots for $n = 4$ individual sample values. For comparisons between conditions, two-tailed Student's *t*-tests or Mann–Whitney tests are used, \*$p < 0.05$, \*\*$p < 0.01$, \*\*\*$p < 0.001$, ns for nonsignificant. Ladder of 100 base pairs (bp).

each specific primers for *Ace2* (forward 5′–CTACAGGCCCTTCAGCAAAG–3′; reverse 5′–TGCCCAGAGCCTAGAGTTGT–3′; product size of 204 bp), *Tmprss2* (forward 5′-ACTGACCTCCTCATGCTGCT-3′; reverse 5′-TGACGATGTTGAGG CTTGC-3′; product size of 225 bp) and *Gapdh* (forward 5′-AACTTTGGCATTGTGG AAGG-3′; reverse 5′-ACACATTGGGGGTAGGAACA-3′; product size of 223 bp). Primers were designed using the Primer3plus resource[32]. Amplification was done with 1.0 units of DNA Polymerase (GoTaq® DNA Polymera; Promega) using a thermocycler (Veriti™; Applied Biosystems) programmed at 95 °C 30 s, 55 °C 30 s, and 68 °C 45 s for 30 cycles. Visualization of the amplification products was done with ethidium bromide on a 3% electrophoresis gel and sizes were assessed (Bench Top 100 bp DNA Ladder; Promega). Negative controls were performed with H$_2$O or by omitting the reverse transcription phase. The RT-PCR semi-quantitative analyses of *Ace2* and *Tmprss2* were calculated according to the intensities detected under the curves[72] (ImageJ; 1.53a) relatively expressed to *Gapdh*.

*qRT-PCR investigations.* For quantitative RT-PCR (qRT-PCR), amplifications of *Ace2* and *Tmprss2*, as well as *Gapdh* as housekeeping gene and H$_2$O for internal control, were performed with a Real-Time PCR system detector (7500 Fast Real-Time PCR System; Applied Biosystems). All reactions were performed in duplicates with a final volume of 20 µl, containing 1× SYBR® Green enzyme (Fast SYBR Green Master Mix; Applied Biosystems), 800 nM of primers and 3 µl of cDNA and by using universal and fast PCR cycling conditions. A threshold line of 0.15 was applied to compare the different C$_T$ (cycle threshold) for further analysis. The RNA-fold change was then calculated using the comparative $2^{(-\Delta\Delta CT)}$ method[73], normalized to *Gapdh*.

**Protein expression profile by immunohistochemistry.** Floating immunohistochemistry procedures were performed to precisely localize protein expression in the different chemosensory epithelia[31,32]. Accordingly, and prior to tissue isolation, mouse heads were chemically fixated in 4% paraformaldehyde (PAF 4%, pH 7.4). After 24 h of fixation, chemosensory epithelia were rinsed in PBS and included in low melting 4% agar. Serial coronal slices of 80–120 µm were generated with a vibroslicer (VT1200S; Leica) and collected in ice-cold PBS. Slices were then selected under a stereomicroscope (M165 FC; Leica) based on their general morphology and/or GFP expression. Selected slices were then blocked for 3 h at room temperature in a PBS solution containing 5% normal donkey serum (NDS; Jackson ImmunoResearch) and 1% of non-ionic detergent (Triton® X-100; Fluka). A double and indirect immunostaining approach using specific primary antibodies was then used to simultaneously localize ACE2 expression with other marker proteins[14,15]. The primary antibodies used were directed against ACE2 (Goat anti-ACE2; PA5-47488; Invitrogen; 1:40), TMPRSS2 (Rabbit anti-TMPRSS2; ab109131; Abcam; 1:200), CK18 (Rabbit anti-CK18; PA5-14263; Invitrogen; 1:50), CK5 (Rabbit anti-CK5; ab52635; Abcam; 1:160), SOX2 (Rabbit anti-SOX2; PA1-094; Invitrogen; 1:200), GUST (Rabbit anti-G$_\alpha$ gust; sc-395; Santa Cruz Biotechnology; 1:250), CNGA2 (Rabbit anti-CNGA2; APC-045; Alomone Laboratories; 1:200), S100B (Rabbit anti-S100β; ab41548; Abcam; 1:500), pGCG (Rabbit anti-PGCG; PGCG-

701AP; FabGennix; 1:250) and γTUB (Rabbit anti-gamma Tubulin; ab179503; Abcam; 1:250). Detection of primary antibodies was performed using fluorochrome-conjugated secondary antibodies against goat (Alexa Fluor Plus 647-conjugated, donkey anti-Goat; A32849; Invitrogen; 1:200) and rabbit (Cy3-conjugated, donkey anti-Rabbit; 711-165-152; Jackson ImmunoResearch; 1:200). Slices were then rinsed in 1% NDS solution; nuclei were counterstained with DAPI thanks to the antifade mounting medium (Vectashield®; H-1200; Vector Labs). Control experiments were performed by omitting primary antibodies and ran in parallel on C57BL/6 mice. Observations were made by LED-fluorescence microscopy (EVOS M5000; Invitrogen) and acquisitions performed with confocal microscopy (SP5; Leica). Maximum projections, reconstructions and colocalization analysis were made with computer assistance (IMARIS 6.3; Bitplane).

**Protein expression profile by western-blot.** Samples for western-blot analysis[73] were generated with chemosensory epithelia obtained from 2 to 6 mice (with a comparable sex ratio). Protein extractions were homogenized in RIPA lysis buffer (Tris 50 mM pH 7.2, NaCl 150 mM, NP40 1%, SDS 0.1%, Na-deoxycholate 0.5%) implemented with proteases inhibitor (Pepstatine A, Aprotinin, Leupeptin hemi-sulfate, PMSF; Merck) and a phosphatase inhibitor cocktail (ThermoFisher Scientific) under high-speed tissue disruption (TissueLyser II; Qiagen). After centrifugation, the supernatants were collected and total protein concentration were assessed according to BCA protein assay kit (Pierce BCA Protein Assay Kit; Thermo Scientific). Then, 25 µg of proteins of each sample were separated on a 10% Tris-acetate gel and electrophoretically transferred to membranes (Amersham Protran 0.2 µm NC; GE Healthcare Life science). Membranes were then incubated with the primary antibody against ACE2 (Goat anti-ACE2; PA5-47488; Invitrogen; 1:500) or ACTIN (Rabbit anti-Actin; A2066; Merck; 1:2500). Appropriate Horseradish Peroxidase-conjugated secondary antibodies were used (Donkey anti-goat, 705-035-003, Jackson ImmunoResearch, 1:4000; Goat anti-rabbit, 111-035-003, Jackson ImmunoResearch, 1:10000) and detected by chemiluminescence (SuperSignal West Pico PLUS Chemiluminescent Substrate; Thermo Scientific) and subsequently acquired (FusionSolo; Viber). The western-blot semi-quantitative analysis of ACE2 was calculated according to the intensities detected under the curves[72] (ImageJ; 1.53a) relatively expressed to ACTIN.

**Statistics and reproducibility.** Statistical analysis, bar graphs, and their corresponding aligned dot plots were computed with GraphPad Prism 8.4.3. Values of independent experiments from distinct samples are expressed as mean ± standard error of the mean (SEM). Sample size was determined on the basis of pilot experiments and according to previously reported publications done in the field. For histological experiments, the observed variability between individual acquisitions, at a given age, was modest and a minimum standard of a triplicate was thus necessary. Comparisons were unpaired and performed with the two-tailed Student's *t*-tests with Welch's correction in the case of non-respect of the

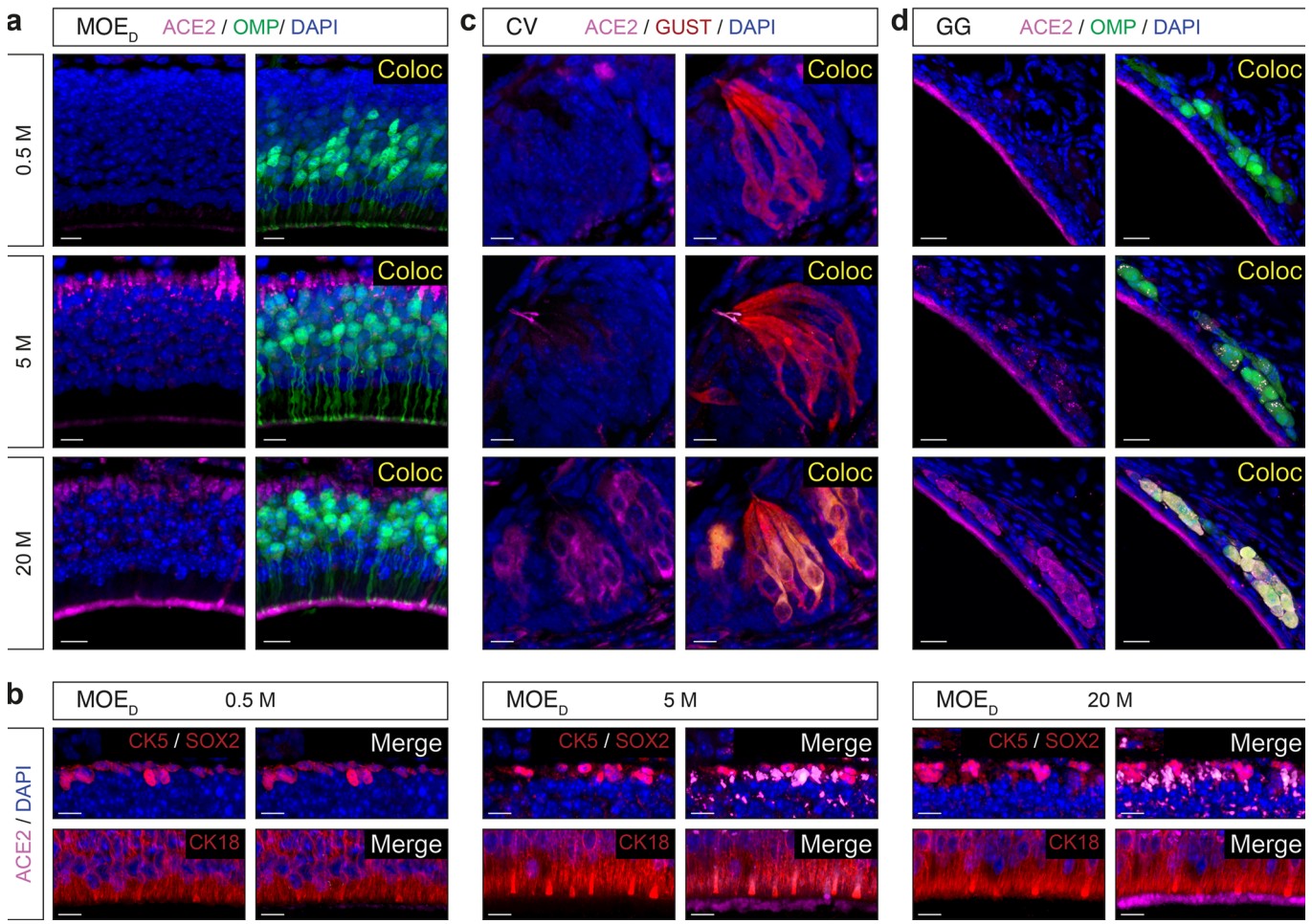

**Fig. 9 Age-dependent ACE2 expression in the MOE_D, CV, and GG.** Assessment of ACE2 expression (in pink) in chemosensory systems by immunohistochemistry at the indicated age in months (M). **a** A gradual increase in ACE2 expression is observed in the apical and basal cell layer of the MOE_D. **b** Age-dependent increase of expression of ACE in the MOE_D is not associated to an increased number of basal (CK5/SOX2+; upper part of panels) or sustentacular (CK18+; lower part of panels) cells. **c** In taste buds of the CV, ACE2 is sequentially expressed in the taste pore microvilli and in the sensory somas. **d** In the GG region, apical keratinocytes constantly express ACE2 while this expression is progressively observed in GG neurons (punctiform to somatic staining). Colocalization (Coloc, in yellow) in sensory cells of ACE2 with OMP (**a**, **d**) or GUST (**c**) markers are highlighted. Colocalization between red and pink signals is highlighted in light gray (**b**). Nuclei are counterstained with Dapi (DAPI, in blue). Representative protein expression profile obtained from heterozygous OMP-GFP mice of 0.5, 5 and 20 months old are indicated. Scale bars are 10 µm (**a–c**) and 20 µm (**d**).

homoscedasticity (Fisher $F$-tests). Mann–Whitney tests were applied in absence of Normality (Shapiro–Wilk test). Significance levels are indicated as follows for: $*p < 0.05$; $**p < 0.01$; $***p < 0.001$; ns for nonsignificant.

**Reporting summary**. Further information on research design is available in the Nature Research Reporting Summary linked to this article.

## Data availability
All data and materials used in the analysis are available in the main text, in the Supplementary Information and in the Supplementary Data 1 file.

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

## Acknowledgements

We thank M. Auberson, F. Durussel for biochemistry advices and E. Corset for technical support; B. Rossier for his thoughtful discussions on the manuscript. This work was supported by the Department of Biomedical Sciences, University of Lausanne, and by the Swiss National Science Foundation Grant 310030_185161 (to M.-C.B.).

## Author contributions

J.B. and M.-C.B designed the project. J.B., A.C.L., D.W., S.B., A.dV., and C.V. carried out experimental procedures. All authors discussed the results and analyzed data. J.B. and M.-C.B. wrote the manuscript.

## Competing interests

The authors declare no competing interests.
