## [Peer Review File · Communications Biology]

Reviewers' Comments:

Reviewer #1:

Remarks to the Author:

In this manuscript, Julien Brechbühl et al., characterized age-dependent expression of genes (ACE2 and TMPRSS2) required for SARS-CoV-2 entry to cells in multiple chemosensory organs, including ones which have not been analyzed previously. The examinations with immunohistochemistry were comprehensive. The high quality of images is very impressive. They identified the dynamic distribution in cells as animals age, i.e., from non-sensory to both non- and sensing cells in mice. Confirmation of an age-dependent expression of Ace2 and Tmprss2 in more than one chemosensory organ provides an interpretation of the age-dependent anosmia-ageusia symptoms in COVID patients. The manuscript was well written.

There are some concerns that need to be addressed:

1. Considering Fungiform papillae take over anterior 2/3 area of the tongue, it is important to analyze ACE2 and TMPSS2 expression in anterior tongue tissue. Such data were not mentioned in this manuscript.
2. The use of red color is concerning. It is not easily distinguished from magenta in figures. Suggest to try other options, e.g., light grey.
3. Age-dependence of the ACE2 and TMPSS2 expression is the strongest point made in this article. It would be great to have analyses in a quantitative manner.
4. In line 37-39, the conclusion sentence "Our results not only clarify human viral-induced sensory symptoms but also propose new investigative perspectives based on ACE2-humanized mouse models." is beyond the support of mouse data presented in this manuscript.
5. Line 208, the subtitle of "SARS-CoV-2 entry site expression in mouse sensory systems" needs to be specific, e.g., chemosensory systems.

Reviewer #2:

Remarks to the Author:

In this work, Brechbühl and colleagues performed a careful and thorough analysis of the expression of ACE2 and TMPRSS2 in five different mouse chemosensory systems: main olfactory epithelium (MOE), Grueneberg ganglion (GG), circumvallate papilla (CV), vomeronasal organ (VNO) and Septal organ (SO). The MOE, GG and CV showed higher expression levels of ACE2 in RT-PCR experiments, and were further analyzed regarding ACE2 and TMPRSS2 protein expression by immunohistochemistry. The authors found that in the MOE these proteins are coexpressed in the Sustentacular cells, basal (progenitor) cells and Bowmans glands, but not in the sensory neurons, as already initially described by other groups (Brann, D. H. et al. *Science advances* 6,doi:10.1126/sciadv.abc5801 (2020); Fodoulia, L. et al., *iScience*. 2020 Dec 18; 23(12): 101839). Differently from the MOE, the authors found that in the GG and CVs, the sensory cells express the virus 'receptors', indicating that these cells can be direct targets of the virus. Finally, the authors show that expression of ACE2 and TMPSSR2 varies depending on the animal's age.

The experiments are well executed and the results raise interesting considerations in terms of how SARS-Cov2 infects chemosensory cells. However, the following points should be addressed:

- 1- Figs. 2A and 3A show extensive staining for ACE2 in the apical region of the MOE. Authors mention that sustentacular cell show apical staining for ACE2 (Fig. 3A). Is this ACE2 staining in the SC's microvilli? There is no mention in the text about this. The apical region of the OE also contains different types of microvillous cells. Studies using transcriptomic analysis have actually suggested that these microvillous cells may express ACE2 (Fodoulia, L. et al., *iScience*. 2020 Dec 18; 23(12): 101839). Can the authors distinguish these cells in their experiments?
- 2- It is hard to discriminate red (SOX2) and pink (ACE2) colors in figures 3b and 3c. In Fig 3b, it does not look like ACE2 is coexpressed with CK5 (HB cells), as stated in the text. The thin layer of cells containing the HB cells seems to be lying underneath the ACE2 positive GB cells. It looks like ACE2 is

coexpressed with Sox2 (GB cells).

3- Line 205: authors write 'systems. Moreover, we bring here evidences of the presence of viral entry sites in chemosensory neurons'. It is important to stress here that the GG neurons are not typical chemosensory neurons, since as the authors themselves mention, they show a double cellular affiliation (neuronal and fibroblastic).

4- Figure 8 shows an interesting finding: ACE2 expression varies depending on the animal's age. This could explain why sometimes results are not congruent between different experiments from different groups. They could also explain why older animals would be more susceptible to infection than younger ones. For Figure 8A, it would have been important to have used markers for the sustentacular cells and for the progenitor cells, since these are the cells co expressing ACE2. Without this, it is hard to tell whether increase in expression with age is due to increase in the number of cells, or due to increase in ACE2 expression in the existing cells. This point is actually stated in the article's title: 'Age-dependent appearance of SARS-CoV-2 entry cells in mouse chemosensory systems reflects COVID-19 anosmia-ageusia symptoms.' Do the cells appear with age or does ACE2 get expressed with age?

5- Discussion could be extended to include two important related aspects of the work:

a) It has been shown that interferon can induce expression of ACE2, so variations in ACE2 expression can be caused by other factors than age; Ziegler, et al (2020). SARSCoV-2 Receptor ACE2 Is an Interferon-Stimulated Gene in Human Airway Epithelial Cells and Is Detected in Specific Cell Subsets across Tissues. *Cell*, 181(1016–1035), e1019. <https://doi.org/10.1016/j.cell.2020.04.035>

b) Mice do not get infected by SARS COV2, because their ACE2 is different from the human one and does not bind the virus spike protein. This is why hACE2 transgenic mice and other modified mice have been used, but in these cases, expression of hACE2 is different from the natural condition. The Syrian hamster seems to be a more physiological alternative model for SARS Cov2 infection experiments.

6- For the same reasons above, one would not expect that mouse GG neurons or taste cells would be direct targets for the virus.

7- Authors should discuss how previous single cell transcriptomic data reconciles with their data in the case of taste cells (For example: Wang, Z. et al. SARS-CoV-2 Receptor ACE2 Is Enriched in a Subpopulation of Mouse Tongue Epithelial Cells in Nongustatory Papillae but Not in Taste Buds or Embryonic Oral Epithelium *ACS Pharmacol Transl Sci* 3, 749-758.)

Reviewer #3:

None

Responses to reviewers:

Reviewer #1:

We would like to thank the reviewer for her/his encouraging comments. We truly appreciate her/his insightful review of our manuscript. We have now addressed all of her/his questions and suggestions and we believe that taking them into account significantly strengthen our revised manuscript.

In summary, in order to take into account, the comments from all reviewers and from the editor, we have now modified the manuscript, the figures and the material and methods accordingly. Here below, please find a list of the main changes:

- We have analyzed the expression of ACE2 and TMPRSS2 in the fungiform and foliate papillae present in the anterior tongue tissue by RT-PCR, qRT-PCR (revised Figure 1 and new Supplementary Figure 1) and by immunohistochemistry (new Supplementary Figures 3 and 4).
- We have quantified the age-dependent expression of the SARS-CoV-2 entry sites by RT-PCR, qRT-PCR and Western-blot (new Figure 8 and new Supplementary figures 6 and 7).
- We have now used markers for the sustentacular and progenitor cells to verify which cells express ACE2 in the dorsal part of the main olfactory epithelium (MOE_D) and address better the age-dependence observed (new panel b, revised Figure 9).
- The colocalization of the red and magenta fluorescent signals appears now in light grey. It helps contrasting better the two fluorescent signals in all adapted figures where red and magenta colors are used together.
- We have now performed additional immunohistochemistry experiments to better describe the microvillar localization of the apical ACE2 signal in the dorsal MOE (new Supplementary Figure 2b).
- We have modified our main text and discussion section to include the important notions and references on the “*interferon-Stimulated Gene*”, “*Syrian hamster*”, and “*single cell transcriptomic data of taste cells*”.

We hope that these changes will allow a better understanding and interpretation of our results. We look forward to the reviewer response and we will be glad to respond to any further questions and comments that she/he may have.

Please find here the responses to her/his specific comments:

1) Considering Fungiform papillae take over anterior 2/3 area of the tongue, it is important to analyze ACE2 and TMPRSS2 expression in anterior tongue tissue. Such data were not mentioned in this manuscript.

Answer: We thank the reviewer for this very relevant comment. We agree with her/him that Fungiform papillae are the most abundant papillae in the tongue and their analysis is therefore essential for the overall pertinence of the manuscript. Accordingly, we have now analyzed the expression of ACE2 and TMPRSS2 in the Fungiform papillae (Fu) present in the anterior tongue tissue by RT-PCR, qRT-PCR (revised Figure 1 and new Supplementary Figure 1) and by immunohistochemistry (new Supplementary Figure 3). We have also performed these analyses on Foliate papillae (Fo; new Supplementary Figure 4). Interestingly, the expression of *Ace2* and *Tmprss2* in Fu and Fo is the same as the one we previously reported in the Circumvallate papillae (CV) as we found a high level of *Ace2* and low level of *Tmprss2* compared to their expression detected in the MOE (revised Figure 1 and new Supplementary Figure 1). Moreover, ACE2 and TMPRSS2 localization in sensory cells composing the Fu, Fo taste buds was also similar as the one of the CV sensory taste cells as we found them in the microvilli of the taste buds, at the level of the taste pores (new Supplementary Figures 3 and 4). We have now mentioned these important results in the main text of our manuscript.

2) The use of red color is concerning. It is not easily distinguished from magenta in figures. Suggest to try other options, e.g., light grey.

Answer: We would like to thank the reviewer for her/his suggestion. To take into account remarks from all reviewers, we have now used light grey to show colocalization of the fluorescent red and magenta signals to improve the contrast. The light grey indeed allows to highlight all pixels that are shared and to conserve the initial intensities of both signals. Accordingly, we have now adapted **all** figures and figure-panels where red and magenta colors are used together. As an example, please find below the comparison between the original Figure 5b / 7c and their revised versions.

3) Age-dependence of the ACE2 and TMPSS2 expression is the strongest point made in this article. It would be great to have analyses in a quantitative manner.

Answer: We would like to thank the reviewer for her/his positive comment. We agree with the reviewer on the importance of these quantitative analysis and we have now performed these additional experiments both at the RNA and protein levels. For that, we first quantified and confirmed the age-dependent expression of the SARS-CoV-2 entry sites by RT-PCR and qRT-PCR analysis in the MOE_D, in the CV and in the Grueneberg ganglion (GG; new Figure 8 and Supplementary Figure 6). At the RNA level, this age-dependence was particularly striking for *Ace2*. Accordingly, we have now also confirmed our immunohistochemical analysis of ACE2 (revised Figure 9) by Western-blot experiments (Supplementary Figure 7). We consider that with these new results, our original findings concerning the age-dependence are now strongly reinforced.

New Figure 8

New Supplementary Figure 7

Finally, we have adapted our main text (*lines 223-241*):

“We considered a potential age-dependent correlation⁴⁷⁻⁵⁰ and thus we next challenged the expression of *Ace2* and *Tmprss2* transcripts through the different chemosensory subsystems using mice of chosen ages (Fig. 8 and Supplementary Fig. 6). Using both RT-PCR and qRT-PCR experiments, we confirmed our assumption at the RNA level as we observed and quantified a significant increase of both *Ace2* and *Tmprss2* transcripts with age in the MOED (Fig. 8a, b and Supplementary Fig. 6a), in the CV (Fig. 8c, d and Supplementary Fig. 6b) and in the GG (Fig. 8e, f and Supplementary Fig. 6c). As this age-dependence was particularly striking for *Ace2*, we next focused on its expression at the protein level (Fig. 9 and Supplementary Fig. 7). After a first validation of this age-dependent trend in a series of Western-blot quantifications (Supplementary Fig. 7a-c), we subsequently undertook a precise immunohistochemical localizations of ACE2

expression in the different chemosensory subsystems (Fig. 9). Accordingly, we observed we first an intense variation of ACE2 its expression in the MOED (Fig. 9a). Indeed, in young mice, this expression was restricted to the apical region, and then gradually intensifies with age (Fig. 9a). At the basal level, it appears to be absent first and only shows up in adult mice (Fig. 9a). Moreover, these variations were directly associated with an increase in ACE2 expression in existing cells. The sustentacular and basal cells are present at all ages of interest (Fig. 9b)”.

4) In line 37-39, the conclusion sentence “Our results not only clarify human viral-induced sensory symptoms but also propose new investigative perspectives based on ACE2-humanized mouse models.” is beyond the support of mouse data presented in this manuscript.

Answer: We agree with the reviewer and we have now modified our conclusion accordingly (*lines 38-41*):

~~“Our results not only clarify human viral-induced sensory symptoms but also propose new investigative perspectives based on ACE2-humanized mouse models.~~ Our results propose a new interpretation of the human viral-induced sensory symptoms and give investigative perspectives on animal models”.

5) Line 208, the subtitle of “SARS-CoV-2 entry site expression in mouse sensory systems” needs to be specific, e.g., chemosensory systems.

Answer: We would like to thank the reviewer for this remark and apologize for the lack of clarity. As suggested by the reviewer, we have now modified this subtitle (*lines 217-218*):

“SARS-CoV-2 entry sites expression in mouse chemosensory systems is age-dependent”.

Reviewer #2:

We would like to thank the reviewer for her/his encouraging comments. We truly appreciate her/his insightful review of our manuscript. We have now addressed all of her/his questions and suggestions and we believe that taking them into account significantly strengthen our revised manuscript.

In summary, in order to take into account, the comments from all reviewers and from the editor, we have now modified the manuscript, the figures and the material and methods accordingly. Here below, please find a list of the main changes:

- We have analyzed the expression of ACE2 and TMPRSS2 in the fungiform and foliate papillae present in the anterior tongue tissue by RT-PCR, qRT-PCR (revised Figure 1 and new Supplementary Figure 1) and by immunohistochemistry (new Supplementary Figures 3 and 4).
- We have quantified the age-dependent expression of the SARS-CoV-2 entry sites by RT-PCR, qRT-PCR and Western-blot (new Figure 8 and new Supplementary figures 6 and 7).
- We have now used markers for the sustentacular and progenitor cells to verify which cells express ACE2 in the dorsal part of the main olfactory epithelium (MOE_D) and address better the age-dependence observed (new panel b, revised Figure 9).
- The colocalization of the red and magenta fluorescent signals appears now in light grey. It helps contrasting better the two fluorescent signals in all adapted figures where red and magenta colors are used together.
- We have now performed additional immunohistochemistry experiments to better describe the microvillar localization of the apical ACE2 signal in the dorsal MOE (new Supplementary Figure 2b).
- We have modified our main text and discussion section to include the important notions and references on the “*interferon-Stimulated Gene*”, “*Syrian hamster*”, and “*single cell transcriptomic data of taste cells*”.

We hope that these changes will allow a better understanding and interpretation of our results. We look forward to the reviewer response and we will be glad to respond to any further questions and comments that she/he may have.

Please find here the responses to her/his specific comments:

*1) Figs. 2A and 3A show extensive staining for ACE2 in the apical region of the MOE. Authors mention that sustentacular cell show apical staining for ACE2 (Fig. 3A). Is this ACE2 staining in the SC's microvilli? There is no mention in the text about this. The apical region of the OE also contains different types of microvillous cells. Studies using transcriptomic analysis have actually suggested that these microvillous cells may express ACE2 (Fodoulian, L. et al., *iScience*. 2020 Dec18; 23(12): 101839). Can the authors distinguish these cells in their experiments?*

Answer: We would like to thank the reviewer for this important comment and apologize for the lack of clarity. We agree with her/him that with immunohistochemical approach, the discrimination between the different types of microvilli and/or cilia that are present in the apical region of the MOE is difficult due to the cellular density. To avoid any overinterpretation of our results, we indeed initially chose to describe that CK18 positive cells were harboring ACE2 (Figure 3a), as it is the only information that we can had with our immunohistochemical approach.

To better challenge the origin of the apical ACE2 signal found in the MOE_D, we have now performed an additional immunohistochemistry experiment adapted from previously established histological approaches (Fodoulian, et al., *iScience*. 2020; Brann, et al., *sciadv*. 2020). For that, we have used a specific olfactory cilia marker, the CNGA2 and we found its signal to be localized above the ACE2 signal (new panel b of the revised Supplementary Figure 2).

New Supplementary Figure 2b

We have now adapted our description accordingly in the manuscript (*lines 136-142*) and we consider that this additional information indeed reinforces our initial description:

“Focusing on the identification of these ACE2-expressing cells (Fig. 3), we first confirmed that the cytokeratin 18 (CK18)-positive sustentacular cells^{16,21}, operating as supporting cells for

the olfactory neurons, were indeed harboring, at their luminal surface, the observed ACE2 protein (Fig. 3a). Moreover, we found ACE2 apical expression to be localized below the olfactory sensory cilia expressing the cyclic nucleotide-gated channel alpha 2 subunits (CNGA2; Supplementary Fig. 2b), supporting its previously reported sustentacular microvilli affiliation^{14,15}.

2) It is hard to discriminate red (SOX2) and pink (ACE2) colors in figures 3b and 3c. In Fig 3b, it does not look like ACE2 is coexpressed with CK5 (HB cells), as stated in the text. The thin layer of cells containing the HB cells seems to be lying underneath the ACE2 positive GB cells. It looks like ACE2 is coexpressed with Sox2 (GB cells).

Answer: We would like to thank the reviewer for expressing these concerns. We agree with her/him that the discrimination between red and pink colors is difficult. To take into account remarks from all reviewers, we have now used light grey to show colocalization of the fluorescent red and pink signals to improve the contrast. The light grey indeed allows to highlight all pixels that are shared and to conserve the initial intensities of both signals. Accordingly, we have now adapted **all** figures and figure-panels where red and pink colors are used together. As an example, please find below the comparison between the original Figure 3b and 3c and their revised version. Interestingly, we found that this allowed a clearer visualization of the CK5 signal, which predominantly appears underneath the ACE2 positive GB cells as a thin perinuclear layer. Moreover, and as requested by the reviewer, we have now clarified this peculiar expression feature in our manuscript (*lines 142-147*):

“We next established the multipotency characteristics of the ACE2-expressing basal cells, as they also expressed two proteins specific to their identity of stem cells, the perinuclear cytokeratin 5 (CK5; Fig. 3b) and, predominantly, the nuclear transcription factor sex determining region Y-box 2 (SOX2; Fig. 3c), markers of horizontal basal cells (HBCs) or of the pear-shaped globose basal cells (GBCs) respectively”.

3) Line 205: authors write “systems. Moreover, we bring here evidences of the presence of viral entry sites in chemosensory neurons”. It is important to stress here that the GG neurons are not typical chemosensory neurons, since as the authors themselves mention, they show a double cellular affiliation (neuronal and fibroblastic).

Answer: We thank the reviewer for sharing her/his concern. We have now modified our description accordingly (lines 213-215):

“Moreover, we bring here evidences of the presence of viral entry sites in *chemosensory neurons* a particular population of chemosensory cells, the olfactory GG neurons”.

4) Figure 8 shows an interesting finding: ACE2 expression varies depending on the animal’s age. This could explain why sometimes results are not congruent between different experiments from different groups. They could also explain why older animals would be more susceptible to infection than younger ones. For Figure 8A, it would have been important to have used markers for the sustentacular cells and for the progenitor cells, since these are the cells co expressing ACE2.

Without this, it is hard to tell whether increase in expression with age is due to increase in the number of cells, or due to increase in ACE2 expression in the existing cells. This point is actually stated in the article's title: "Age-dependent appearance of SARS-CoV-2 entry cells in mouse chemosensory systems reflects COVID-19 anosmia-ageusia symptoms". Do the cells appear with age or does ACE2 get expressed with age?

Answer: We thank the reviewer for this positive and relevant comment. We agree with the reviewer on the importance of these control experiments that we have now performed. We found that both progenitor cells (CK5/SOX2 stainings) and sustentacular cells (CK18 stainings) are present at all the investigated ages in the MOE_D. Thus, the observed increase in expression of ACE2 with age is not due to an increase in the number of cells expressing ACE2, but rather due to an increase of ACE2 expression itself in the already existing cells. This finding is particularly important. We have therefore added a new panel **b** for the Figure 9.

New Figure 9b

We have also modified our main text accordingly and we propose a new title for our manuscript to avoid any confusion:

(lines 237-241) Main text: "Indeed, in young mice, this expression was restricted to the apical region, and then gradually intensifies with age (Fig. 9a). At the basal level, it appears to be absent first and only shows up in adult mice (Fig. 9a). Moreover, these variations were directly associated with an increase in ACE2 expression itself in existing cells, sustentacular and basal cells being present at all ages of interest (Fig. 9b)".

*(lines 2-3) Title: "Age-dependent appearance of SARS-CoV-2 entry **cells** sites in mouse chemosensory systems reflects COVID-19 anosmia-ageusia symptoms".*

5) Discussion could be extended to include two important related aspects of the work:

a) It has been shown that interferon can induce expression of ACE2, so variations in ACE2 expression can be caused by other factors than age; Ziegler, et al (2020). *SARSCoV-2 Receptor ACE2 Is an Interferon-Stimulated Gene in Human Airway Epithelial Cells and Is Detected in Specific Cell Subsets across Tissues. Cell, 181(1016–1035), e1019. <https://doi.org/10.1016/j.cell.2020.04.035>*

b) Mice do not get infected by SARS COV2, because their ACE2 is different from the human one and does not bind the virus spike protein. This is why hACE2 transgenic mice and other modified mice have been used, but in these cases, expression of hACE2 is different from the natural condition. The Syrian hamster seems to be a more physiological alternative model for SARS Cov2 infection experiments.

Answer: We agree with the reviewer that these important notions were lacking from our original discussion. Their addition is a plus for the general clarity and readability of our manuscript. Accordingly, we have now added new references (Ziegler, et al., *Cell*. (2020); Rosa, et al., *Commun Biol*. (2021); Shigemura, et al., *Nutrients*. (2019). de Melo, *Sci Transl Med*. (2021)). We have mentioned, in our discussion, the “interferon-Stimulated Gene” and the “Syrian hamster – hACE2” points, (lines 287-306):

“Thus, SARS-CoV-2 could therefore also directly target these sensory cells in aged mice expressing, for example, the human form of ACE2 (hACE2) under the mouse Ace2 promotor^{45,57} and, possibly, in aged humans⁴², altering them and consequently disabling the taste sensory ability leading to ageusia symptoms. Thus, the use of animal models such as mice sensitized to the virus^{45,57-60} or Syrian hamsters that are naturally compatible with SARS-CoV-2 infections^{19,21,61}, appears to be a promising strategy but the age of the animals has to be carefully considered and older animals should preferentially be used (from 9 months of age for mice). To reinforce the importance of this notion, previous elegant single cell transcriptomic analysis indeed reported only limited expression of Ace2 in taste cells of young and embryonic mice¹⁷. We reported here a similar observation both at the RNA and protein levels in young mice and we completed this description by looking at ACE2 expression in aged mice. Thus, we found that the age of the animal directly impacts the interpretations of the results obtained in SARS-CoV-2 studies^{17,42,62}. Interestingly, based on our results, additional studies performed on the lower respiratory airways could show whether a similar increased ACE2 expression, as a function of

age occurs not only in respiratory ciliated cells^{49,63} but also in basal cells which could therefore contribute, in association with other factors such as interferon-stimulation^{64,65}, to the differences in pulmonary infectiousness observed between young and older people⁶⁶”.

6) For the same reasons above, one would not expect that mouse GG neurons or taste cells would be direct targets for the virus.

Answer: Similarly, as above, we have now adapted our text and mentioned the importance of considering GG neurons and taste cells from either hACE2 or hamster models for studying SARS-CoV-2 thanks to their virus compatibility.

7) Authors should discuss how previous single cell transcriptomic data reconciles with their data in the case of taste cells (For example: Wang, Z. et al. SARS-CoV-2 Receptor ACE2 Is Enriched in a Subpopulation of Mouse Tongue Epithelial Cells in Non gustatory Papillae but Not in Taste Buds or Embryonic Oral Epithelium ACS Pharmacol Transl Sci 3, 749-758.).

Answer: We thank the reviewer for this integrative comment. We indeed consider that our results strengthen previous reports and complete the global knowledge about the SARS-CoV-2 entry mechanistics. We have now modified our discussion accordingly and quoted this important reference, (lines 287-301):

“Thus, SARS-CoV-2 could therefore also directly target these sensory cells in aged mice expressing, for example, the human form of ACE2 (hACE2) under the mouse Ace2 promotor^{45,57} and, possibly, in aged humans⁴², altering them and consequently disabling the taste sensory ability leading to ageusia symptoms. Thus, the use of animal models such as mice sensitized to the virus^{45,57-60} or Syrian hamsters that are naturally compatible with SARS-CoV-2 infections^{19,21,61}, appears to be a promising strategy but the age of the animals has to be carefully considered and older animals should preferentially be used (from 9 months of age for mice). To reinforce the importance of this notion, previous elegant single cell transcriptomic analysis indeed reported only limited expression of Ace2 in taste cells of young and embryonic mice¹⁷. We reported here a similar observation both at the RNA and protein levels in young mice and we completed this description by looking at ACE2 expression in aged mice. Thus, we found that the age of the animal directly impacts the interpretations of the results obtained in SARS-CoV-2 studies^{17,42,62}”.

Reviewers' Comments:

Reviewer #1:

Remarks to the Author:

The authors did a great job in responding to Reviewers' comments and addressing all concerns.
No more concerns.

Reviewer #2:

None

Reviewer #3:

Remarks to the Author:

Authors replies TO THE COMMENTS AND EREVISED THE MANUSCRIPT ACCORDINGLY. NO MORE
COMMENTS

Responses to reviewers:

Reviewer #1: *The authors did a great job in responding to Reviewers' comments and addressing all concerns. No more concerns.*

Reviewer #3: *Authors replies TO THE COMMENTS AND EREVISED THE MANUSCRIPT ACCORDINGLY. NO MORE COMMENTS*

Answer: We would like to thank the reviewers for the approval of our manuscript. We truly appreciated their respective insightful reviews and for their significant contributions.